# Clinicopathological Features of Kidney Injury Related to Immune Checkpoint Inhibitors: A Systematic Review

**DOI:** 10.3390/jcm12041349

**Published:** 2023-02-08

**Authors:** Ling-Yi Xu, Hai-Ya Zhao, Xiao-Juan Yu, Jin-Wei Wang, Xi-Zi Zheng, Lei Jiang, Su-Xia Wang, Gang Liu, Li Yang

**Affiliations:** 1Renal Division, Department of Medicine, Institute of Nephrology, Peking University First Hospital, Peking University, Beijing 100034, China; 2Renal Pathology Center, Institute of Nephrology, Peking University First Hospital, Beijing 100034, China; 3Key Laboratory of Renal Disease, Ministry of Health of China, Beijing 100034, China; 4Key Laboratory of CKD Prevention and Treatment, Ministry of Education of China, Beijing 100034, China; 5Research Units of Diagnosis and Treatment of Immune-Mediated Kidney Diseases, Chinese Academy of Medical Sciences, Beijing 100034, China; 6Eight-Year-Program, Grade 2019, Health Science Center, Peking University, Beijing 100191, China; 7Laboratory of Electron Microscopy, Pathological Center, Peking University First Hospital, Beijing 100034, China

**Keywords:** acute kidney injury, immune checkpoint inhibitors, immune-related adverse events, pathological features

## Abstract

(1) Background: Despite increasing recognition of immune checkpoint inhibitors (ICIs) and kidney immune-related adverse events (IRAEs), no large-sample studies have assessed the pathological characteristics and outcomes of biopsy-proven kidney IRAEs. (2) Methods: We comprehensively searched PubMed, Embase, Web of Science, and Cochrane for case reports, case series, and cohort studies for patients with biopsy-proven kidney IRAEs. All data were used to describe pathological characteristics and outcomes, and individual-level data from case reports and case series were pooled to analyze risk factors associated with different pathologies and prognoses. (3) Results: In total, 384 patients from 127 studies were enrolled. Most patients were treated with PD-1/PD-L1 inhibitors (76%), and 95% presented with acute kidney disease (AKD). Acute tubulointerstitial nephritis/acute interstitial nephritis (ATIN/AIN) was the most common pathologic type (72%). Most patients (89%) received steroid therapy, and 14% (42/292) required RRT. Among AKD patients, 17% (48/287) had no kidney recovery. Analyses of pooled individual-level data from 221 patients revealed that male sex, older age, and proton pump inhibitor (PPI) exposure were associated with ICI-associated ATIN/AIN. Patients with glomerular injury had an increased risk of tumor progression (OR 2.975; 95% CI, 1.176, 7.527; *p* = 0.021), and ATIN/AIN posed a decreased risk of death (OR 0.164; 95% CI, 0.057, 0.473; *p* = 0.001). (4) Conclusions: We provide the first systematic review of biopsy-proven ICI-kidney IRAEs of interest to clinicians. Oncologists and nephrologists should consider obtaining a kidney biopsy when clinically indicated.

## 1. Introduction

Immune checkpoint inhibitors (ICIs) are among the most promising therapeutic approaches in the fight against cancer, and have revolutionized the treatment of different types of advanced cancer in recent years. An increasing number of studies have shown the remarkable efficacy of ICIs in multiple solid and hematologic cancers, and a considerable number of patients benefit from ICI therapy with improved survival [1,2,3]. ICIs can interrupt coinhibitory signaling pathways by blocking immune-regulatory inhibitory receptors, such as cytotoxic T-lymphocyte-associated antigen-4 (CTLA-4) and programmed cell death protein-1 (PD-1), thereby mitigating T-cell suppression and promoting T-cell activation and proliferation. Eventually, they can induce a potent T-cell-mediated antitumor immune response [4,5]. However, upregulation of the T-cell-mediated immune response has been associated with a wide spectrum of immune-related adverse events (IRAEs), affecting various organs [6].

Recently, an increasing number of studies and guidelines have started to focus on kidney IRAEs associated with ICI-associated kidney IRAEs (ICI-kidney IRAEs), and have attempted to summarize the clinical features and treatment of kidney injury. According to the available cohort studies, among patients with ICI treatments, the incidence of acute kidney injury (AKI) is 0.4% to 5.0% [7,8,9], yet non-AKI kidney involvement has rarely been described, and kidney biopsy had not been conducted in most patients. As ICIs are increasingly available and recommended for clinical use, the number of kidney IRAEs may be more prevalent globally [10]. Based on case reports and case series with biopsy-proven ICI-kidney IRAEs, the most common pathological lesions are acute tubular necrosis (ATN) and acute tubulointerstitial nephritis (ATIN), followed by minimal change disease (MCD), vasculitis, glomerulonephritis (GN), and IgA nephropathy (IgAN). Due to the limited sample size of each study, it is difficult to describe the distribution and comparison of the features and prognosis of kidney pathologies from IRAEs.

In this study, we systematically reviewed all published studies reporting cases with biopsy-proven ICI-kidney IRAEs. Altogether, 384 patients from 127 studies were included, and the clinical features, kidney pathologies, treatments, and outcomes of kidney IRAEs were described. The potential risk factors associated with different kidney pathologies, kidney outcomes, and oncologic outcomes were further analyzed based on the pooled individual data.

## 2. Materials and Methods

This study was performed following the Preferred Reporting Items for Systematic Reviews and Meta-Analyses (PRISMA) guidelines [11]. The study protocol has been registered with PROSPREO, number CRD CRD42022289336 (https://www.crd.york.ac.uk/prospero/ (accessed on 21 January 2022)).

### 2.1. Search Strategy and Selection Criteria

A comprehensive literature search was performed in the PubMed, Embase, Web of Science, and Cochrane databases until November 2021. The key search terms included immune checkpoint inhibitors, kidney-adverse events, and some terms relevant to “kidney disease”. Full details on the search strategy and search terms are provided in Additional File: Appendix A.

We enrolled studies with the following criteria: (1) adult patients (aged >18 years) and (2) patients who received kidney biopsy due to kidney-adverse events after ICI therapy. Exclusion criteria were as follows: (1) lack of detailed pathologic information for kidney-adverse events, (2) lack of detailed clinical information for kidney-adverse events, (3) reviews, preclinical studies, animal studies, and systematic review without exhaustive data, (4) non-English language articles, and (5) publications for which the full text could not be found. Only the most recent and complete report was included if duplicate publications or overlapping study populations were identified. Two reviewers (LYX and HYZ) independently screened all the titles and abstracts to identify potentially relevant studies, followed by full-text screening to determine the final eligibility of the studies. Disagreements were resolved through discussion or by the decision of a third reviewer (XJY), if necessary.

### 2.2. Data Extraction and Quality Assessment

Relevant information was independently extracted from each included study by two authors (LYX and HYZ). The data extraction included the study type, study size, demographic characteristics, comorbidities, tumor type, ICI treatment information (i.e., type, dose, duration from first treatment to the occurrence of kidney IRAEs), biochemical characteristics, histopathologic renal findings, information on kidney-adverse events (i.e., characteristics, stage, treatment regimens) and outcomes (i.e., tumor outcome, kidney outcome, rechallenge).

Patients with biopsy-proven kidney IRAEs were classified into acute kidney disease (AKD) and non-AKD groups, according to the reported changes in kidney function. AKD was defined if AKI or AKD was clearly described in the original articles. Where not otherwise stated in the original articles, AKD was defined by peak serum creatinine (SCr)/baseline SCr ≥1.5 or an eGFR decrease of ≥35% [12]. AKD stage was defined by the level of peak SCr/baseline SCr and the usage of renal replacement treatment (RRT), according to Acute Disease Quality Initiative (ADQI) guidelines [12]. Kidney outcomes were judged mainly based on the relevant specific reports in the article. If no relevant reports were available, the kidney outcomes were estimated by the decline in SCr or proteinuria levels. Detailed definitions are shown in Appendix A. Tumor outcomes were defined by the specific reported tumor outcomes after relevant intervention had been applied for kidney IRAE in the original article, and were classified into four categories: complete remission (CR), partial remission (PR), stable disease (SD), and progressive disease (PD). The disease control rate (DCR) was calculated using (CR+PR+SD)/(CR+PR+SD+PD).

The kidney pathologies were extracted from the relevant reports in the original article and were classified into four categories: ATIN/acute interstitial nephritis (AIN), ATN, glomerular diseases, and systematic diseases. Glomerular diseases consist of podocyte diseases [focal segmental glomerulosclerosis (FSGS), MCD], IgAN, membranous nephropathy (MN), and GN (crescent GN, C3 GN, immune-mediated GN, and unclassed GN). Crescent GN refers to glomerulonephritis with crescents that cannot be identified as a specific disease, such as vasculitis and anti-glomerular basement membrane disease (anti-GBM disease). Systematic diseases consist of vasculitis/ANCA-related vasculitis, thrombotic microangiopathy (TMA), kidney graft rejection, renal amyloidosis (AA), lupus nephritis, and anti-GBM disease.

The methodological quality of case reports or case series was evaluated according to the Mayo Evidence-Based Practice Centre tool using four domains (selection, ascertainment, causality, and reporting) [13]. The Newcastle–Ottawa Scale was used to evaluate the quality and risk of bias of the case–control and cohort studies [14]. All quality assessments were completed by two reviewers (LYX and HYZ) independently, and disagreements were resolved through discussion.

### 2.3. Statistical Analyses

The data from the cohort study, case series, and case report were used to describe the baseline characteristics, pathologic type, treatment, and kidney and tumor outcomes by descriptive statistics. The individual-level data from case reports and case series were pooled to analyze factors associated with different pathologies, kidney recovery, tumor outcome, and death by univariate and multivariable logistic regression. Complete available-case analysis was used based on different outcomes, and missing data were not imputed. Categorical variables were expressed as frequencies, and continuous variables were expressed as the mean ± SD or medians (interquartile range), depending on whether the data were normally distributed. *T* tests and nonparametric tests were used for between-group comparisons based on whether a normal distribution existed. The chi-square test was used for categorical variables. Multivariable logistic regression was additionally performed with the variables significant at *p* < 0.10 in univariate logistic regression. The odds ratio (OR) was used to represent the strength of the association. All analyses were performed using SPSS 26.0 software (SPSS Inc., Chicago, IL, USA), and *p* < 0.05 was considered statistically significant.

## 3. Results

### 3.1. Clinical Characteristics of Patients with ICI-Kidney IRAEs

The search strategy identified 2272 records in total, and 127 studies were included after screening and eligibility assessment. Altogether, 384 patients with ICI-associated, biopsy-proven kidney IRAEs were cumulatively enrolled, including 221 patients from the 122 case reports and case series and 163 patients from five observational cohort studies (summarized in Appendix A). Quality assessment forms are provided in Appendix A. The flow chart of the study enrollment process is shown in Figure 1. As shown in Table 1, of all the patients, 68% (261/384) were male, with hypertension in 50% (151/305) and diabetes in 14% (44/307). The most frequent tumor types were melanoma (150/384, 39%) and lung cancer (110/384, 29%). Among the 372 patients who had detailed information on ICI regimens, PD-1/PD-L1 inhibitors were mostly used (281/372, 76%), CTLA-4 inhibitors were taken by 22 (6%), and combination therapy with PD-1/PD-L1 plus CTLA-4 was prescribed for 69 (19%). Of the 282 patients with detailed antitumor therapy data, 12% (33/282) received combination treatment with other antitumor therapies, and the most frequent combination treatment was chemotherapy (20/282, 7%).

Among these 384 patients with biopsy-proven ICI-kidney IRAEs, 95% (363/384) presented with AKD, and 5% (21/384) had proteinuria with normal kidney function. Various kidney histopathologic lesions were observed. ATIN/AIN was the most common pathologic type (277/384, 72%), followed by glomerular diseases (54/384, 14%), systematic diseases (42/384, 11%), and ATN (29/384, 8%). Podocyte diseases (23/384, 6%), GN (15/384, 4%), vasculitis (18/384, 5%), and TMA (8/384, 2%) were the most common lesions of glomerular diseases and systematic diseases. Details are shown in Table 2. Among patients with detailed AKD stage, the majority (146/242, 60%) were stage 3, and a total of 14% (42/292) of patients required RRT, among 292 AKD patients with detailed information on RRT. Among 340 patients reporting extrarenal IRAEs, 34% (117/340) had simultaneous ICI-related IRAEs of the kidney and of other extrarenal organs, and in particular, skin (36/340, 11%), gastrointestinal (30/284, 11%), and endocrine systems (22/284, 7%) were the most common extrarenal IRAEs, with 56 patients lacking gastrointestinal and endocrine IRAE data.

### 3.2. Treatment and Outcome of Patients with ICI-Kidney IRAEs

ICI therapy was discontinued in most patients after kidney injury (322/336, 96%). The majority of patients (342/384, 89%) received steroid therapy for ICI-associated IRAEs, with a median oral steroid dose of 1.0 (0.8, 1.0) mg/kg/d, while 26% (76/289) received intravenous steroids, and among the cases reporting intravenous steroid dose, 72% (31/43) received pulse steroid therapy at a dose of 701 ± 262 mg/ds. Of 351 patients reporting immunosuppressant agents, 14% (41/351) were treated with additional immunosuppressant agents for severe IRAEs, relapse of kidney injury, or kidney transplant patients. Eight patients (ATN in five, immune GN in one, anti-GBM disease in one, and AA in one) were treated with immunosuppressant agents without steroids. The remaining 11% (44/384) of patients did not receive steroid or immunosuppressant therapy due to mild symptoms (Table 3).

Altogether, 311 patients had reports on their renal outcome, with 24 patients having data only for complete recovery. Of the AKD patients, 42% (129/311) obtained complete recovery, 43% (123/287) had partial recovery, and 17% (48/287) had no recovery. Of the 36 patients for whom there was detailed information on RRT, only 39% discontinued RRT. Of patients with proteinuria and normal renal function, 80% (16/20) achieved proteinuria remission. Of the 93 patients with information on their tumor outcomes, 44% (41/93) had tumor progression. Altogether, 14% of patients (36/252) in the original articles died.

### 3.3. Pooled Individual Analysis of Patients with ICI-Associated Kidney IRAEs

To explore the potential factors that may be relevant to the different histopathology of kidney IRAEs and to the patients’ outcomes, we further analyzed the pooled individual-level data of the 221 patients enrolled from the case reports and case series. Factors identified as significant in univariate analysis (Appendix A) were entered into multivariable analysis. We found that male sex (OR 0.321; 95% CI 0.118, 0.875; *p* = 0.026), older age (OR 1.055; 95% CI, 1.016, 1.097; *p* = 0.006) and PPI treatment (OR 3.620; 95% CI, 1.357, 9.658; *p* = 0.010) were associated with ATIN/AIN in multivariable analysis, while older age was found to be inversely associated with glomerular diseases (OR 0.933; 95% CI, 0.895, 0.973; *p* = 0.001) after adjusting for confounders (shown in Figure 2). Multivariable logistic regression analysis was not performed for ATN and systematic disease, as few factors were significant in the univariate analysis.

We next explored the factors associated with kidney recovery in AKD patients. As shown in Figure 3, the use of steroids was independently associated with kidney recovery (OR 9.429, 95% CI, 1.823, 48.779; *p* = 0.007), whereas systematic diseases (OR 0.119, 95% CI, 0.038, 0.376; *p* < 0.001) and the use of RRT (OR 0.111, 95% CI, 0.033, 0.374; *p* < 0.001) had significant inverse associations with kidney recovery. Potential risk factors for tumor progression and death were also analyzed. As shown in Appendix A, there were no significant associations between tumor progression or death and receiving steroids (including pulse steroid therapy), immunosuppressants, or discontinued ICI treatment. Patients with glomerular diseases had an increased risk of tumor progression (OR 2.975; 95% CI, 1.176, 7.527; *p* = 0.021) in multivariable analysis. Male sex (OR 3.136; 95% CI, 1.026, 9.583; *p* = 0.045), ATIN/AIN (OR 0.180; 95% CI, 0.071, 0.455; *p* < 0.001), systematic diseases (OR 3.123; 95% CI, 1.293, 7.539; *p* = 0.011), and no recovery of kidney function (OR 4.455; 95% CI, 1.554, 12.767; *p* = 0.004) were related to death in univariate analysis. After adjusting for confounding factors, patients with ATIN/AIN had a decreased risk for death (OR 0.164; 95% CI, 0.057, 0.473; *p* = 0.001), and those with hematologic cancers (OR 13.342, 95% CI, 1.032, 172.512; *p* = 0.047) had a 13.3-fold risk of death compared to that in patients with melanoma (Figure 4).

### 3.4. Immunohistochemical Staining of Biopsied Kidney Tissues

Twenty-four patients from 17 case reports [15,16,17,18,19,20,21,22,23,24,25,26,27,28,29,30,31] and 3 case series [32,33,34] had information on the immunohistochemistry (IHC) staining of kidney infiltrating cells. All 24 patients presented with ATIN, with one case having concomitant glomerulonephritis (not specified). Kidney infiltration of T lymphocytes (CD3+) and B lymphocytes (CD20+) was detected in 100% (21/21) and 80% (8/10) of patients, respectively. Of the 10 patients who had kidney sections stained for both anti-CD3 and anti-CD20, the number of T cells was found to be significantly higher than that of B cells. CD4+ T-cell (18/18) and CD8+ T-cell (20/20) infiltration was detected in all patients, with 69% (9/13) of patients having CD4+ T-cell dominance and 31% (4/13) having CD8+ T-cell dominance. Macrophage (CD68+) infiltration was found in all the tested patients (7/7).

IHC staining for PD-L1 and PD-1 was performed in 23 patients from eight case reports [15,16,19,20,23,26,27,29] and one cohort study [35], with ATIN in 17 patients and ATN in 6 patients. Positive PD-L1 staining was detected in tubular epithelial cells in 15/17 (88%) patients with ATIN, but not in the ATN patients (*p* < 0.001). Slight focal PD-1 staining in inflammatory cells was detected in 16/18 (89%) patients, with no difference between patients with ATIN (10/12, 83%) and those with ATN (6/6, 100%) (*p* = 1.000).

## 4. Discussion

With the expanding application of ICIs, kidney IRAEs have gained increasing attention. This is the first systematic review on published biopsy-proven ICI-kidney IRAEs, which allowed us to have a more comprehensive understanding of the clinical features, pathologies, treatments, and outcomes of ICI-kidney IRAEs. Meanwhile, with cumulative data, we were able to identify the factors associated with different kidney pathotypes and outcomes, which may help guide diagnostic and treatment strategies.

ICI-associated IRAEs have been reported to be related to age and sex. For example, female patients exhibit an increased risk of ICI-associated IRAEs [36], especially endocrine [36], skin [37], and thyroid gland [38] IRAEs. Older patients have an increased risk of lung [39] and skin [40] IRAEs, while younger patients have an increased risk of liver [39] and endocrine [40] IRAEs. To date, there have been no reports on the relevance of age and sex to ICI-related kidney IRAEs. In the current study, we found that compared to male patients, female patients had an increased risk of ATIN/AIN; moreover, older patients had an increased risk of ATIN/AIN, while younger patients were more likely to have glomerular diseases. The reasons for the discrepancies in age and sex and their relevance to ICI-IRAEs are thus far undefined. One hypothesis suggests that the immune system undergoes a wide range of changes during aging; this process, called “immunosenescence” [41], may contribute to the difference in the immune system between older and younger patients. Therefore, ICIs may affect immune responses differently in older and younger patients, leading to different types of IRAEs [42]. More research is needed to further explore the discrepancies and the related mechanisms of age- and sex-associated ICI-kidney IRAEs.

Treatment of severe ICI-kidney IRAEs with steroids is recommended by the current guidelines [43]. In the current study, we found that most of the reported patients with biopsy-proven ICI-kidney IRAEs received steroid therapy (89%) with a median oral steroid dose of 1.0 (0.8, 1.0) mg/kg/d, and most patients received ICI therapy (96%). The use of steroids was independently associated with kidney recovery, which is consistent with previous studies [44,45]. It is interesting to note that receiving steroids and immunosuppressants and halting ICI treatment were not associated with an increased risk of tumor progression or death, which indicates that immunosuppressive therapies (steroids or immunosuppressants) and the discontinuation of ICI therapy may not influence the tumor response to ICIs or the risk of death; thus, administering more aggressive steroid therapy and withholding ICI treatment may improve kidney outcomes in patients with severe kidney IRAEs. However, the current evidence remains too limited to support this finding, and further studies with a higher strength of evidence are warranted.

Data on risk factors for death in patients with ICI-kidney IRAEs are sparse, to our knowledge, as different risk factors have only been reported in two studies. Shruti et al. found that only a lower baseline eGFR was associated with a higher risk of death in 405 patients diagnosed with ICI-associated AKI [46]. Meanwhile, different factors were identified in a multicenter study conducted by Cortazar et al. [44], and it was reported that patient kidney recovery was associated with lower mortality in 138 patients with ICI-associated AKI. However, in the current study, we found that ATIN/AIN was an independent protective factor for death in patients with ICI-kidney IRAEs. Unlike previous studies, our study population consisted of patients with ICI-kidney IRAEs, including non-AKI kidney disease and ICI-associated AKI, and we included variables related to pathology in univariate and multivariable analyses. This might have contributed to the difference in results from those of the previous studies. The lower mortality associated with ATIN/AIN from ICI-kidney IRAEs may reflect that patients with suspected ICI-associated ATIN/AIN require closer attention, which highlights the potential benefits of early recognition and treatment for ATIN/AIN.

Studies have been conducted to explore the mechanisms of ICI-related kidney IRAEs. Analysis of the IHC staining results in the case reports showed that predominant T-cell infiltration was detected in all the available tested kidney biopsies, with the majority of the patients presenting with CD4+ T-cell dominance (69%). Moreover, HLA-DR expression was detected in renal tubular epithelial cells [31]. These results support the current view that ICI-induced T-cell hyperactivation mediates kidney injury. CTLA-4/PD-1 signaling failure suggests that certain intrinsic kidney antigens might become targets for aberrantly activated T cells [47]. B cells were detected in the biopsied kidney sections in 80% of patients with ICI-kidney IRAEs, which indicates a potential role of B cells in kidney IRAEs development. Previous studies have shown that augmented T-cell–B-cell interactions may result in autoantibody production, due to increased T-cell activation with ICIs [48]. PD-L1 expression was detected in tubular epithelial cells in most of the patients with anti-PD-L1 IHC staining (65.2%), which suggests a potential mechanism in which the anti-PD-L1 monoclonal antibody directly binds to renal tubular cells and mediates tubular injury [49]. Finally, the use of PPIs was found to independently increase the risk of ICI-associated ATIN/AIN in the current analysis, which is consistent with previous studies [44,49]. It is possible that PPI may latently prime T cells as haptens, and that ICI further activates these T cells and leads to loss of tolerance [50]. Moreover, the possibility cannot be entirely ruled out that ATIN/AIN may be directly associated with PPIs instead of ICIs. Currently, it cannot be determined whether ICIs, in combination with PPIs, may increase the risk of ICI-kidney IRAEs due to a lack of quality data from large studies.

Our study is the first systematic review of this topic to date and provides key insights into ICI-related kidney IRAEs. Nevertheless, there were still several limitations in our study. As mentioned above, the current review predominantly consisted of case reports or case series, and there was a small number of prospective studies. Lower levels of evidence and a higher risk of reporting and selection bias may thus exist. Nonuniform reporting of patient descriptors led to complete clinical data not being available in a proportion of patients, particularly in retrospective observational studies. Small sample sizes and missing information were present when some outcomes were analyzed, particularly regarding the tumor outcome and treatment dose. In the enrolled studies, especially the case reports, there may have been substantial variation in the hospital settings, physician level, follow-up time and other factors, which might have contributed to heterogeneity. Further heterogeneity was difficult to assess in our study because some of the associated factors were not reported in the original articles. With an increasing appearance of new studies [51,52,53], our understanding of ICI-kidney IRAEs will be deepened gradually. Further studies are needed to explore the related mechanism and help better manage the disease.

## 5. Conclusions

In conclusion, we provide the first systematic review data for biopsy-proven ICI-kidney IRAEs of interest to clinicians, including a comprehensive description of the pathological spectrum of the various patterns, risk factors for the development of different pathologies, and risk factors associated with prognosis. Oncologists and nephrologists should be aware that different pathologies affect different kidney and tumor outcomes and should consider obtaining a kidney biopsy when it is clinically indicated.

## Figures and Tables

**Figure 1 jcm-12-01349-f001:**
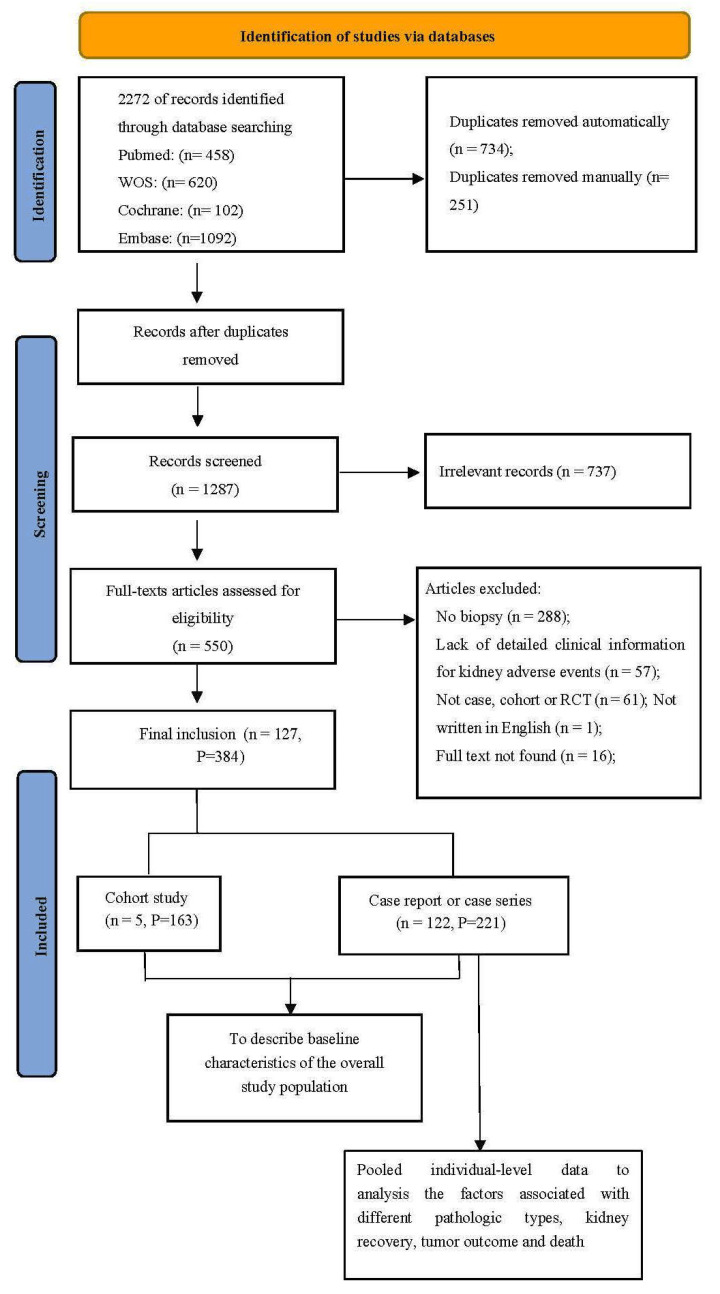
PRISMA 2009 Flow Diagram. “n” represents the number of studies; “P” represents patients.

**Figure 2 jcm-12-01349-f002:**
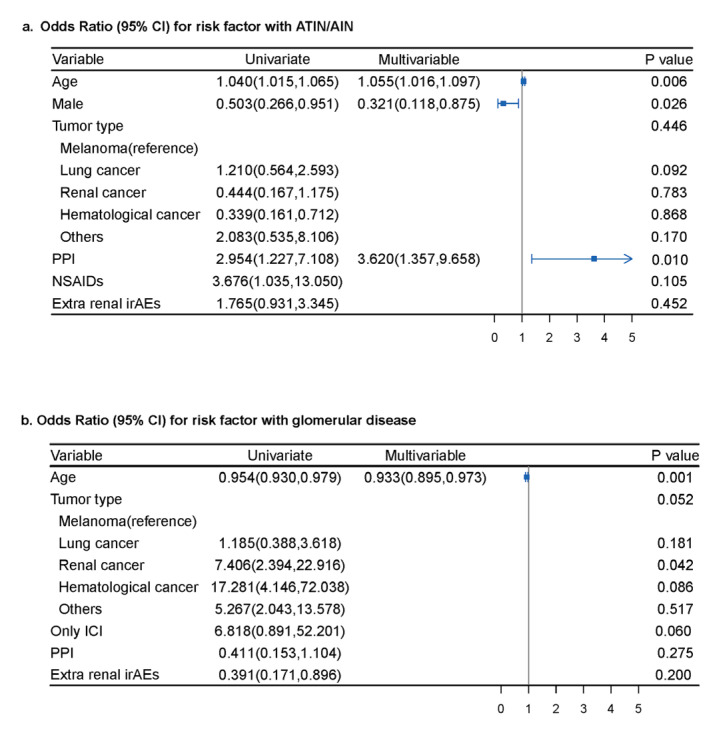
Factors associated with the occurrence of renal adverse events with ATIN/AIN and glomerular disease. (**a**) Male sex, older age and PPI treatment were associated with ATIN/AIN in multivariable analysis; (**b**) older age was inversely associated with glomerular diseases after adjusting for confounders. ATIN/AIN: acute tubulointerstitial nephritis/acute interstitial nephritis; PPI: proton pump inhibitor; ICI: immune checkpoint inhibitor; CI: confidence interval.

**Figure 3 jcm-12-01349-f003:**
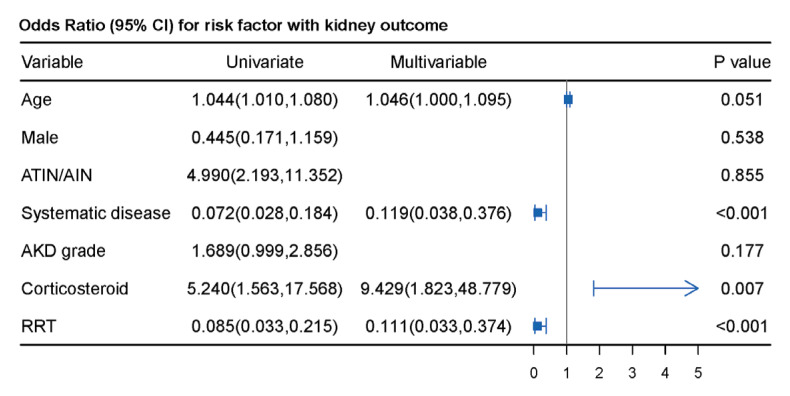
Multiple logistic analysis of factors associated with kidney function recovery in AKD patients. ATIN/AIN: acute tubulointerstitial nephritis/acute interstitial nephritis; AKD: acute kidney disease; CI: confidence interval; RRT: renal replacement treatment.

**Figure 4 jcm-12-01349-f004:**
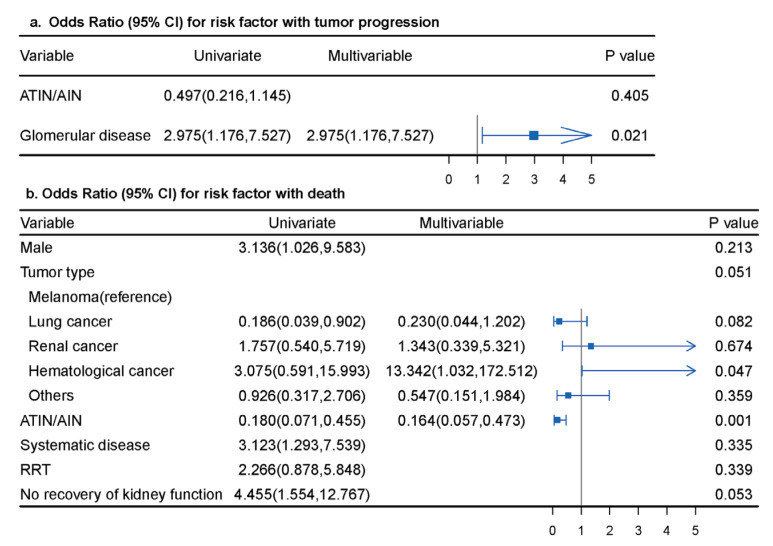
Univariate logistic analysis of factors associated with tumor progression and death. (**a**) Risk factors associated with tumor progression; (**b**) risk factors associated with death. ATIN/AIN: acute tubulointerstitial nephritis/acute interstitial nephritis; CI: confidence interval; RRT: renal replacement treatment.

**Table 1 jcm-12-01349-t001:** Summary of baseline and clinical characteristics of the total group of patients with biopsy-proven kidney IRAEs.

	Total (*n* = 384)	Cohort (*n* = 163)	Case (*n* = 221)
Age (year)	67 (59.73)	NA	67 (59.73)
Male [*n/N*(%)]	261/384 (68.0)	108/163 (66.3)	153/221 (69.2)
Hypertension [*n/N*(%)]	151/305 (49.5)	75/148 (50.7)	76/157 (48.4)
Diabetics [*n/N*(%)]	44/307 (14.3)	17/148 (11.5)	27/159 (17.0)
CKD [*n/N*(%)]	60/219 (27.4)	19/60 (31.7)	41/159 (25.8)
CHD [*n/N*(%)]	20/222 (9.0)	5/63 (7.9)	15/159 (9.4)
Tumor type			
Melanoma [*n*/*N*(%)]	150/384 (39.0)	63/163 (38.7) ^e^	87/221 (39.4)
Lung cancer [*n*/*N*(%)]	110/384 (28.6)	54/163 (33.1) ^e^	56/221 (25.3)
Renal cancer [*n*/*N*(%)]	29/324 (9.0)	8/103 (7.8)	21/221 (9.5)
Hematologic cancer ^a^ [*n*/*N*(%)]	13/324 (4.0)	2/103 (1.9)	11/221 (5.0)
Others ^b^ [*n*/*N*(%)]	57/324 (17.6)	11/103 (10.7)	46/221 (20.8)
ICI type			
PD-1/PD-L1 [n/*N*(%)]	281/372 (75.5)	129/163 (79.1)	152/209 (72.8)
CTLA-4 [*n*/*N*(%)]	22/372 (5.9)	6/163 (3.7)	16/209 (7.7)
Combination ^c^ [n/*N*(%)]	69/372 (18.5)	28/163 (17.2)	41/209 (19.6)
Other treatment			
Only ICI [*n*/*N*(%)]	249/282 (88.3)	94/102 (92.1)	155/180 (86.1)
ICI+ targeted therapy [*n*/*N*(%)]	8/282 (2.8)	1/102 (1.0)	7/180 (3.9)
ICI+ chemotherapy [*n*/*N*(%)]	20/282 (7.1)	5/102 (4.9)	15/180 (8.3)
ICI+VEGF [*n*/*N*(%)]	5/282 (1.8)	2/102 (2.0)	3/180 (1.7)
Interval time (d)	105 (59,210)	NA	105 (59,210)
PPI [*n*/*N*(%)]	123/272 (45.2)	71/137 (51.8)	52/135 (38.5)
NSAIDs [*n*/*N*(%)]	36/213 (16.9)	9/78 (11.5)	27/135 (20.0)
Baseline SCr (mg/dl)	1.0 (0.8,1.2)	NA	1.0 (0.8,1.2)
Peak SCr (mg/dl)	3.9 (2.5,5.9)	NA	3.9 (2.5,5.9)
AKD [*n*/*N*(%)]	363/384 (94.5)	162/163 (99.4)	201/221 (91.0)
AKD grade			
1 grade [*n*/*N*(%)]	25/179 (14.0)	NA	25/179 (14.0)
2 grade [*n*/*N*(%)]	31/179 (17.3)	NA	31/179 (17.3)
3 grade [*n*/*N*(%)]	146/242 (60.3)	23/63 (36.5) ^f^	123/179 (68.7)
Only proteinuria [*n*/*N*(%)]	21/384 (5.5)	1/163 (0.6)	20/221 (9.0)
Extrarenal IRAE [*n*/*N*(%)]	117/340 (34.4)	53/134 (39.6)	64/206 (31.1)
Skin [*n*/*N*(%)]	36/340 (10.0)	14/134 (10.4) ^g^	22/206 (10.7)
Gastrointestinal [*n*/*N*(%)]	30/284 (10.6)	9/78 (11.5)	21/206 (10.2)
Endocrine [*n*/*N*(%)]	22/284 (6.5)	5/78 (6.4)	17/206 (8.3)
Pneumonitis [*n*/*N*(%)]	10/340 (2.9)	3/134 (2.2) ^g^	7/206 (3.4)
Others ^d^ [*n*/*N*(%)]	31/284 (4.6)	1/78 (1.3)	12/206 (5.8)

Abbreviations: NA: not available; IRAE: immune-related adverse events; ICI: immune checkpoint inhibitor; CKD: chronic kidney disease; CHD: coronary heart disease; VEGF: vascular endothelial growth factor; Interval time: time between the first ICI and RAE occurrences; PPI: proton pump inhibiter; NSAIDs: nonsteroidal anti-inflammatory drugs; SCr: serum creatinine; AKD: acute kidney disease. ^a^. Hematologic carcinoma included lymphoma and myelodysplastic syndrome. ^b^. Other carcinomas included osteosarcoma, urothelial carcinoma, pancreatic cancer, colorectal cancer, endometrial cancer, mesothelioma, prostate cancer, breast cancer, liposarcoma, thyroid cancer, liver cancer, adrenal cortex cancer, and ovarian cancer. ^c^. Combination was considered when both immune checkpoint inhibitors were coadministered simultaneously. ^d^. Other extrarenal IRAEs included nervous, oculus, heart, encephalitis, and rheumatic IRAEs. ^e^. One cohort study (Frank B. Cortazar) reported data only for melanoma and lung cancer. ^f^. One cohort study (Alexandre O) reported data only for AKD grade 3. ^g^. One cohort study (Frank B. Cortazar) reported data only for skin and pneumonitis IRAEs.

**Table 2 jcm-12-01349-t002:** Histopathological type of total patients with biopsy-proven kidney IRAEs.

	Total (*n* = 384)	Cohort (*n* = 163)	Case (*n* = 221)
ATIN/AIN [*n*/*N*(%)]	277 (72.3)	134	143 ^a^
ATIN [*n*/*N*(%)]	191 (49.9)	121	70
AIN [*n*/*N*(%)]	86 (22.5)	13	73
Total ATN [*n*/*N*(%)]	29 (7.6)	18	11
ATN alone [*n*/*N*(%)]	23 (6.0)	17	6
ATN with glomerular injury [*n*/*N*(%)]	6 (1.6)	1	5
Glomerular injury [*n*/*N*(%)]	54 (14.1)	8	46 ^b^
Podocyte injury [*n*/*N*(%)]	23 (6.0)	4	19
FSGS [*n*/*N*(%)]	8 (2.1)	1	7
MCD [*n*/*N*(%)]	15 (3.9)	3	12
IgA nephropathy [*n*/*N*(%)]	8 (2.1)	0	8
MN [*n*/*N*(%)]	11 (2.9)	1	10
GN [*n*/*N*(%)]	15 (3.9)	3	12
Crescent GN [*n*/*N*(%)]	2 (0.5)	0	2
C3 GN [*n*/*N*(%)]	3 (0.8)	1	2
Immune-mediated GN [*n*/*N*(%)]	5 (1.3)	0	5
Unclassed GN [*n*/*N*(%)]	5 (1.3)	2	3
Systematic disease [*n*/*N*(%)]	42 (10.9)	3	39 ^c^
Vasculitis/ANCA vasculitis [*n*/*N*(%)]	18 (4.7)	2	16
Anti-GBM disease [*n*/*N*(%)]	4 (1.0)	1	3
TMA [*n*/*N*(%)]	8 (2.1)	0	8
AA amyloidosis [*n*/*N*(%)]	5 (1.3)	0	5
Renal graft rejection [*n*/*N*(%)]	7 (1.8)	0	7
Lupus nephritis [*n*/*N*(%)]	1 (0.3)	0	1

Abbreviations: ATN: acute tubular necrosis; ATIN: acute tubulointerstitial nephritis; AIN: acute interstitial nephritis; MCD: minimal change disease; FSGS: focal segmental glomerulosclerosis; TMA: thrombotic microangiopathy; MN: membranous nephropathy; GN: glomerulonephritis; AA: amyloidosis protein A; anti-GBM disease: anti-glomerular basement membrane disease. ^a^. Ten patients had ATIN/AIN with glomerular injury, and three patients had ATIN/AIN with systematic disease in renal biopsy. ^b^. Three patients presented with two types of glomerular injury. ^c^. One patient presented with two types of systematic disease.

**Table 3 jcm-12-01349-t003:** Treatment and outcome of total patients with biopsy-proven kidney IRAEs.

	Total (*n* = 384)	Cohort (*n* = 163)	Case (*n* = 221)
Corticosteroid [*n*/*N*(%)]	342/384 (89.1)	138/163 (84.7)	204/221 (92.3)
Intravenous corticosteroid [*n*/*N*(%)]	76/289 (26.3)	18/88 (20.5)	58/201 (28.9)
Dose (mg/d)	500 (225,1000)	NA	500 (225–1000)
Pulse steroid therapy [*n*/*N*(%)]	31/43 (72.1)	NA	31/43 (72.1)
Dose (mg/d)	701 ± 262	NA	701 ± 262
Oral corticosteroid (mg/kg/d)	1.0 (0.8,1.0)	NA	1.0 (0.8–1.0)
Low-dose (<0.5 mg/kg/d)	18/136 (13.2)	NA	18/136 (13.2)
Moderate-dose (0.5–1.0 mg/kg/d)	99/136 (72.8)	NA	99/136 (72.8)
High-dose (>1.0 mg/kg/d)	19/136 (14.0)	NA	19/136 (14.0)
Immunosuppressants [*n*/*N*(%)]	49/351 (14.0)	7/148 (4.7)	42/203 (20.7) ^a^
Infliximab [*n*/*N*(%)]	12/351 (3.4)	NA	12/203 (5.6)
Rituximab [*n*/*N*(%)]	15/351 (4.3)	NA	15/203 (7.4)
MMF [*n*/*N*(%)]	11/351 (3.1)	NA	11/203 (5.4)
Others [*n*/*N*(%)]	7/351 (2.0)	NA	7/203 (3.5)
Discontinued ICI [*n*/*N*(%)]	322/336 (95.8)	139/145 (95.9)	183/191 (95.8)
RRT [*n*/*N*(%)]	42/292 (14.4)	7/90 (7.8)	35/202 (17.3)
Disruption of RRT [*n*/*N*(%)]	14/36 (38.9)	1/2 (50.0)	13/34 (38.2)
Recovery SCr level (mg/dl)	1.5 (1.1,1.8)	NA	1.5 (1.1,1.8)
Protein remission [*n*/*N*(%)]	16/20 (80.0)	NA	16/20 (80.0)
Renal function recovery			
Complete recovery [*n*/*N*(%)]	129/311 (41.5)	56/143 (39.2) ^b^	73/168 (43.5)
Partial recovery [*n*/*N*(%)]	123/287 (42.9)	59/119 (49.6)	64/168 (38.1)
No recovery [*n*/*N*(%)]	48/287 (16.7)	17/119 (14.3)	31/168 (18.5)
Death [*n*/*N*(%)]	36/252 (14.3)	8/90 (8.9)	28/162 (17.3)
Tumor response			
DCR [*n*/*N*(%)]	52/93 (55.9)	NA	52/93 (55.9)
Complete response [*n*/*N*(%)]	14/93 (15.1)	NA	14/93 (15.1)
Partial response [*n*/*N*(%)]	14/93 (15.1)	NA	14/93 (15.1)
Stable [*n*/*N*(%)]	24/93 (25.8)	NA	24/93 (25.8)
Progression [*n*/*N*(%)]	41/93 (44.1)	NA	41/93 (44.1)
Rechallenge [*n*/*N*(%)]	41/190 (21.7)	18/84 (21.4)	23/106 (21.7)
Flare [*n*/*N*(%)]	10/39 (25.6)	1/18 (5.6)	9/21 (42.9)

Abbreviations: NA: not available; IRAE: immune-related adverse events; MMF: mycophenolate mofetil; RRT: renal replacement therapy; SCr: serum creatine; ICI: immune checkpoint inhibitor; DCR: disease control rate. ^a^. Three patients were treated with both immunosuppressants: one patient received rituximab and MMF, and two patients received infliximab and MMF. ^b^. Two cohort studies (Clarissa Cassol and Juliana B) only provided data for complete renal function recovery.

## Data Availability

The data presented in this study are available in the Appendix A.

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
