# Peer review of "Clinicopathological Features of Kidney Injury Related to Immune Checkpoint Inhibitors: A Systematic Review"

_jcm, 2023, doi:10.3390/jcm12041349_

Round 1
Reviewer 1 Report
Dear Author/s
Greetings for you. I appreciate your work and effort.
Immune checkpoint inhibitors (ICIs) and kidney immune-related adverse events (IRAEs) are often overlooked by the clinician. Renal functions should be closely monitored after treatment. When kidney injury detects in these patients, CKD may develop if they are not treated. Tubulointerstitial nephritis is common in these patients. The number of references in the article is too many, it is appropriate to reduce it according to the journal writing rules.
Best regards
Author Response
Reviewer 1#: Immune checkpoint inhibitors (ICIs) and kidney immune-related adverse events (IRAEs) are often overlooked by the clinician. Renal functions should be closely monitored after treatment. When kidney injury detects in these patients, CKD may develop if they are not treated. Tubulointerstitial nephritis is common in these patients.
Suggestion:
- The number of references in the article is too many, it is appropriate to reduce it according to the journal writing rules.
Response: We thank the Reviewer for the suggestion. We have reduced the number of references appropriately and selected the most relevant in the revised manuscript.

Reviewer 2 Report
The authors firstly reported a systematic review for biopsy-proven immune checkpoint inhibitors (ICIs)-kidney immune-related adverse events (IRAEs). This manuscript is very important because we have more opportunities to diagnose kidney-IRAEs than ever before. I think it is worthy for the warrant publication in Journal of Clinical Medicine, but there are some points to think about.
Minor points:
1) The authors should check spelling of search words in No. 7 of Renal adverse events in Supplemental Table 1.
2) The authors should check whether they unified fonts in Figure 1.
3) What is the difference between acute tubulointerstitial nephritis and acute interstitial nephritis?
Author Response
Reviewer 2#: The authors firstly reported a systematic review for biopsy-proven immune checkpoint inhibitors (ICIs)-kidney immune-related adverse events (IRAEs). This manuscript is very important because we have more opportunities to diagnose kidney-IRAEs than ever before. I think it is worthy for the warrant publication in Journal of Clinical Medicine, but there are some points to think about.
We are sincerely thankful for the Reviewer’s comment and are greatly encouraged by the comment about our work being “very important”.
Suggestion:
1.The authors should check spelling of search words in No. 7 of Renal adverse events in Supplemental Table 1.
Response: We thank the Reviewer for the suggestion and apologize for the unclarity of description in the previous manuscript. Supplemental Table 1 showed our search strategy. In No.7, we used the wildcard symbol * for word retrieval to search the literature as comprehensive as possible. For example, "injur*" means the search would find all the words starting with “injur” including injury, injuries, injured and so on. We have added notes under the Supplemental Table 1 for clarity (“* means wildcard symbol, for example, "injur*" means the search would find all the words starting with “injur” including injury, injuries, injured and so on.”).
2.The authors should check whether they unified fonts in Figure 1.
Response: We have unified the fonts in the revised manuscript.
3.What is the difference between acute tubulointerstitial nephritis and acute interstitial nephritis?
Response: We thank the Reviewer for raising this important question. Acute interstitial nephritis (AIN) connotes predominant involvement of the renal interstitium and tubules by inflammatory cells, often with edema or fibrosis and tubular atrophy. Because interstitial nephritis is commonly accompanied by variable tubular damage, the term acute tubulointerstitial nephritis (ATIN) is preferable and is often used interchangeably with AIN [1]. Considering ATIN and AIN are both referred to in the included studies, we used ATIN/AIN as the uniform definition in the manuscript.
[1] J. Charles Jennette. Heptinstall's Pathology of the Kidney (7th ed.) [M]. Lippincott Williams Wilkins, 2014.

Reviewer 3 Report
The authors provide a systematic review of the published studies reporting cases with biopsy-proven ICI-kidney immune-related adverse events. Based on pooled individual data, potential risk factors for various kidney pathologies, kidney outcomes, and oncologic outcomes were further examined.This is an important topic for which a comprehensive review is scarce in the literature. Although there are several typos and the narrative is a kind of fragmentary, I just abide by scientific soundness. The text is well written. I would like to offer the following minor points for consideration by the authors towards the improvement of the manuscript:
1-The English language used throughout the manuscript needs some improvements in terms of style and grammar
2-Since there are many studies in the last year, it would be good to update the search range. Some examples include:
-DOÄ°: 10.3389/fimmu.2022.993622
-DOI: 10.1016/j.ekir.2021.08.021
-DOI: 10.1097/CAD.0000000000001463
- doi: 10.1007/s13730-021-00645-3
3-Please explain why did not you include these studies to review.
-https://doi.org/10.1177/1078155220961553
-DOI: 10.1097/MD.0000000000027546
-DOI: 10.1007/s13730-020-00462-0
Author Response
Reviewer 3#: The authors provide a systematic review of the published studies reporting cases with biopsy-proven ICI-kidney immune-related adverse events. Based on pooled individual data, potential risk factors for various kidney pathologies, kidney outcomes, and oncologic outcomes were further examined. This is an important topic for which a comprehensive review is scarce in the literature. Although there are several typos and the narrative is a kind of fragmentary, I just abide by scientific soundness. The text is well written.
Suggestions:
- The English language used throughout the manuscript needs some improvements in terms of style and grammar.
Response: We thank the Reviewer for the suggestion. The English language has been improved by an English language editing company called AJE.
- Since there are many studies in the last year, it would be good to update the search range. Some examples include: (1) DOÄ°: 10.3389/fimmu.2022.993622, (2) DOI: 10.1016/j.ekir.2021.08.021, (3) DOI: 10.1097/CAD.0000000000001463, (4) DOI: 10.1007/s13730-021-00645-3.
Response: We thank the Reviewer for the valuable suggestion. The initial literature search was conducted until November 2021 in the PubMed, Embase, Web of Science, and Cochrane databases. Following the Reviewer’s suggestion, we have updated the search from November 2021 until January 2023, and found that 29 new articles including 27 case reports and 2 cohort studies fulfilled the inclusion and exclusion criteria. We then performed an initial analyses and found that adding these new studies would not affect the distribution of the disease spectrum or provide additional information on disease mechanism. Due to the 5-day revision deadline, we would not be able to re-perform the thorough analysis of the whole study. We think the existing data in the meta-analysis could be representative for the clinical-pathological spectrum of ICI associated kidney IRAEs to some extent. As pointed out by the Reviewer, new reports on kidney injury related to ICI are continuously appearing, we have added such discussion in the last paragraph of Discussion section (“With an increasing appearance of new studies, our understanding of ICI-kidney IRAEs will be deepened gradually. Further studies are needed to explore the related mechanism and help better manage the disease.”).
3-Please explain why did not you include these studies to review: (1) https://doi.org/10.1177/1078155220961553, (2) DOI: 10.1097/MD.0000000000027546, (3) DOI: 10.1007/s13730-020-00462-0.
Response: We thank the Reviewer for raising this question. According to the inclusion criteria, we only included patients who received kidney biopsy due to kidney adverse events after ICI therapy. Patients in the literature of (1) https://doi.org/10.1177/1078155220961553 and (3) DOI: 10.1007/s13730-020-00462-0 did not have kidney biopsy and therefore were not included in this systematic review. Literature (2) DOI: 10.1097/MD.0000000000027546 had been included in our review, the detailed information of this literature is listed in supplementary Table S4.
